# Triple descent and the two kinds of overfitting: Where & why do they appear?

**Stéphane d'Ascoli**
Department of Physics
École Normale Supérieure
Paris, France
stephane.dascoli@ens.fr

**Levent Sagun**
Facebook AI Research
Paris, France
leventsagun@fb.com

**Giulio Biroli**
Department of Physics
École Normale Supérieure
Paris, France
giulio.biroli@ens.fr

## Abstract

A recent line of research has highlighted the existence of a "double descent" phenomenon in deep learning, whereby increasing the number of training examples $N$ causes the generalization error of neural networks to peak when $N$ is of the same order as the number of parameters $P$. In earlier works, a similar phenomenon was shown to exist in simpler models such as linear regression, where the peak instead occurs when $N$ is equal to the input dimension $D$. Since both peaks coincide with the interpolation threshold, they are often conflated in the litterature. In this paper, we show that despite their apparent similarity, these two scenarios are inherently different. In fact, both peaks can co-exist when neural networks are applied to noisy regression tasks. The relative size of the peaks is then governed by the degree of nonlinearity of the activation function. Building on recent developments in the analysis of random feature models, we provide a theoretical ground for this sample-wise *triple descent*. As shown previously, the *nonlinear peak* at $N = P$ is a true divergence caused by the extreme sensitivity of the output function to both the noise corrupting the labels and the initialization of the random features (or the weights in neural networks). This peak survives in the absence of noise, but can be suppressed by regularization. In contrast, the *linear peak* at $N = D$ is solely due to overfitting the noise in the labels, and forms earlier during training. We show that this peak is implicitly regularized by the nonlinearity, which is why it only becomes salient at high noise and is weakly affected by explicit regularization. Throughout the paper, we compare analytical results obtained in the random feature model with the outcomes of numerical experiments involving deep neural networks.

## Introduction

A few years ago, deep neural networks achieved breakthroughs in a variety of contexts [1, 2, 3, 4]. However, their remarkable generalization abilities have puzzled rigorous understanding [5, 6, 7]: classical learning theory predicts that generalization error should follow a U-shaped curve as the number of parameters $P$ increases, and a monotonous decrease as the number of training examples $N$ increases. Instead, recent developments show that deep neural networks, as well as other machine learning models, exhibit a starkly different behaviour. In the absence of regularization, increasing $P$ and $N$ respectively yields parameter-wise and sample-wise *double descent* curves [8, 9, 10, 11, 12, 13], whereby the generalization error first decreases, then peaks at the *interpolation threshold* (at which point training error vanishes), then decreases monotonically again. This peak[1] was shown to be related

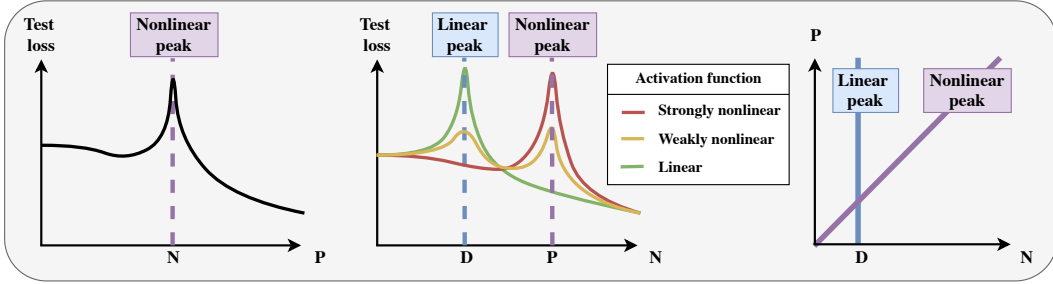

Figure 1: **Left:** The parameter-wise profile of the test loss exhibits double descent, with a peak at $P = N$. **Middle:** The sample-wise profile can, at high noise, exhibit a single peak at $N = P$, a single peak at $N = D$, or a combination of the two (triple descent[2]) depending on the degree of nonlinearity of the activation function. **Right:** Color-coded location of the peaks in the $(P, N)$ phase space.

to a sharp increase in the variance of the estimator [18, 9], and can be suppressed by regularization or ensembling procedures [9, 19, 20].

Although double descent has only recently gained interest in the context of deep learning, a seemingly similar phenomenon has been well-known for several decades for simpler models such as least squares regression [21, 22, 23, 14, 15], and has recently been studied in more detail in an attempt to shed light on the double descent curve observed in deep learning [24, 25, 26, 27]. However, in the context of linear models, the number of parameters $P$ is not a free parameter: it is necessarily equal to the input dimension $D$. The interpolation threshold occurs at $N = D$, and coincides with a peak in the test loss which we refer to as the *linear peak*. For neural networks with nonlinear activations, the interpolation threshold surprisingly becomes independent of $D$ and is instead observed when the number of training examples is of the same order as the total number of training parameters, i.e. $N \sim P$: we refer to the corresponding peak as the *nonlinear peak*.

Somewhere in between these two scenarios lies the case of neural networks with *linear* activations. They have $P > D$ parameters, but only $D$ of them are independent: the interpolation threshold occurs at $N = D$. However, their dynamical behaviour shares some similarities with that of deep nonlinear networks, and their analytical tractability has given them significant attention [28, 29, 6]. A natural question is the following: what would happen for a "quasi-linear" network, e.g. one that uses a sigmoidal activation function with a high saturation plateau? Would the overfitting peak be observed both at $N = D$ and $N = P$, or would it somehow lie in between?

In this work, we unveil the similarities and the differences between the linear and nonlinear peaks. In particular, we address the following questions:

- Are the linear and nonlinear peaks two different phenomena?
- If so, can both be observed simultaneously, and can we differentiate their sources?
- How are they affected by the activation function? Can they both be suppressed by regularizing or ensembling? Do they appear at the same time during training?

**Contribution** In modern neural networks, the double descent phenomenon is mostly studied by increasing the number of parameters $P$ (Fig. 1, left), and more rarely, by increasing the number of training examples $N$ (Fig. 1, middle) [13]. The analysis of linear models is instead performed by varying the ratio $P/N$. By studying the full $(P, N)$ phase space (Fig. 1, right), we disentangle the role of the linear and the nonlinear peaks in modern neural networks, and elucidate the role of the input dimension $D$.

In Sec. 1, we demonstrate that the linear and nonlinear peaks are two different phenomena by showing that they can co-exist in the $(P, N)$ phase space in noisy regression tasks. This leads to a sample-wise *triple descent*, as sketched in Fig. 1. We consider both an analytically tractable model of random features [30] and a more realistic model of neural networks.

In Sec. 2, we provide a theoretical analysis of this phenomenon in the random feature model. We examine the eigenspectrum of random feature Gram matrices and show that whereas the nonlinear

peak is caused by the presence of small eigenvalues [6], the small eigenvalues causing the linear peak gradually disappear when the activation function becomes nonlinear: the linear peak is implicitly regularized by the nonlinearity. Through a bias-variance decomposition of the test loss, we reveal that the linear peak is solely caused by overfitting the noise corrupting the labels, whereas the nonlinear peak is also caused by the variance due to the initialization of the random feature vectors (which plays the role of the initialization of the weights in neural networks).

Finally, in Sec. 3, we present the phenomenological differences which follow from the theoretical analysis. Increasing the degree of nonlinearity of the activation function weakens the linear peak and strengthens the nonlinear peak. We also find that the nonlinear peak can be suppressed by regularizing or ensembling, whereas the linear peak cannot since it is already implicitly regularized. Finally, we note that the nonlinear peak appears much later under gradient descent dynamics than the linear peak, since it is caused by small eigenmodes which are slow to learn.

**Related work**   Various sources of sample-wise non-monotonicity have been observed since the 1990s, from linear regression [14] to simple classification tasks [31, 32]. In the context of adversarial training, [33] shows that increasing $N$ can help or hurt generalization depending on the strength of the adversary. In the non-parametric setting of [34], an upper bound on the test loss is shown to exhibit *multiple descent*, with peaks at each $N = D^i, i \in \mathbb{N}$.

Two concurrent papers also discuss the existence of a triple descent curve, albeit of different nature to ours. On one hand, [19] observes a sample-wise triple descent in a non-isotropic linear regression task. In their setup, the two peaks stem from the block structure of the covariance of the input data, which presents two eigenspaces of different variance; both peaks boil down to what we call "linear peaks". [35] pushed this idea to the extreme by designing the covariance matrix in such a way to make an arbitrary number of linear peaks appear.

On the other hand, [36] presents a parameter-wise triple descent curve in a regression task using the Neural Tangent Kernel of a two-layer network. Here the two peaks stem from the block structure of the covariance of the random feature Gram matrix, which contains a block of linear size in input dimension (features of the second layer, i.e. the ones studied here), and a block of quadratic size (features of the first layer). In this case, both peaks are "nonlinear peaks".

The triple descent curve presented here is of different nature: it stems from the general properties of nonlinear projections, rather than the particular structure chosen for the data [19] or regression kernel [36]. To the best of our knowledge, the disentanglement of linear and nonlinear peaks presented here is novel, and its importance is highlighted by the abundance of papers discussing both kinds of peaks.

On the analytical side, our work directly uses the results for high-dimensional random features models derived in [11, 37] (for the test loss), [38] (for the spectral analysis) and [20] (for the bias-variance decomposition).

**Reproducibility**   We release the code necessary to reproduce the data and figures in this paper publicly at https://github.com/sdascoli/triple-descent-paper.

# 1   Triple descent in the test loss phase space

We compute the $(P, N)$ phase space of the test loss in noisy regression tasks to demonstrate the triple descent phenomenon. We start by introducing the two models which we will study throughout the paper: on the analytical side, the random feature model, and on the numerical side, a teacher-student task involving neural networks trained with gradient descent.

**Dataset**   For both models, the input data $\boldsymbol{X} \in \mathbb{R}^{N \times D}$ consists of $N$ vectors in $D$ dimensions whose elements are drawn i.i.d. from $\mathcal{N}(0, 1)$. For each model, there is an associated label generator $f^\star$ corrupted by additive Gaussian noise: $y = f^\star(\boldsymbol{x}) + \epsilon$, where the noise variance is inversely related to the signal to noise ratio (SNR), $\epsilon \sim \mathcal{N}(0, 1/\text{SNR})$.

## 1.1   Random features regression (RF model)

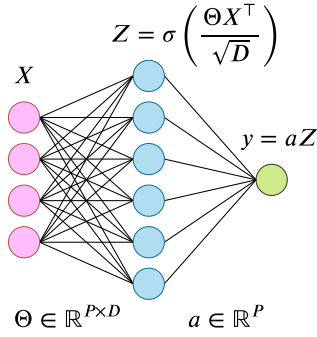

$$Z = \sigma\left(\frac{\Theta X^{\top}}{\sqrt{D}}\right)$$

$X$

$y = aZ$

$\Theta \in \mathbb{R}^{P \times D}$      $a \in \mathbb{R}^{P}$

Figure 2: Illustration of an RF network.

**Model**   We consider the random features (RF) model introduced in [30]. It can be viewed as a two-layer neural network whose first layer is a fixed random matrix $\Theta \in \mathbb{R}^{P \times D}$ containing the $P$ random feature vectors (see Fig. 2)[3]:

$$f(\boldsymbol{x}) = \sum_{i=1}^{P} \boldsymbol{a}_i \sigma\left(\frac{\langle \boldsymbol{\Theta}_i, \boldsymbol{x} \rangle}{\sqrt{D}}\right). \tag{1}$$

$\sigma$ is a pointwise activation function, the choice of which will be of prime importance in the study. The ground truth is a linear model given by $f^{\star}(\boldsymbol{x}) = \langle \boldsymbol{\beta}, \boldsymbol{x} \rangle / \sqrt{D}$. Elements of $\boldsymbol{\Theta}$ and $\boldsymbol{\beta}$ are drawn i.i.d from $\mathcal{N}(0, 1)$.

**Training**   The second layer weights, i.e. the elements of $\boldsymbol{a}$, are calculated via ridge regression with a regularization parameter $\gamma$:

$$\hat{\boldsymbol{a}} = \underset{\boldsymbol{a} \in \mathbb{R}^P}{\arg\min} \left[ \frac{1}{N} \left(\boldsymbol{y} - \boldsymbol{a}\mathbf{Z}^{\top}\right)^2 + \frac{P\gamma}{D}\|\boldsymbol{a}\|_2^2 \right] = \frac{1}{N} \boldsymbol{y}^{\top} \mathbf{Z} \left(\boldsymbol{\Sigma} + \frac{P\gamma}{D}\mathbb{I}_P\right)^{-1} \tag{2}$$

$$\mathbf{Z}_i^{\mu} = \sigma\left(\frac{\langle \boldsymbol{\Theta}_i, \boldsymbol{X}_{\mu} \rangle}{\sqrt{D}}\right) \in \mathbb{R}^{N \times P}, \quad \boldsymbol{\Sigma} = \frac{1}{N} \mathbf{Z}^{\top} \mathbf{Z} \in \mathbb{R}^{P \times P} \tag{3}$$

## 1.2   Teacher-student regression with neural networks (NN model)

**Model**   We consider a teacher-student neural network (NN) framework where a *student* network learns to reproduce the labels of a *teacher* network. The teacher $f^{\star}$ is taken to be an untrained ReLU fully-connected network with 3 layers of weights and 100 nodes per layer. The student $f$ is a fully-connected network with 3 layers of weights and nonlinearity $\sigma$. Both are initialized with the default PyTorch initialization.

**Training**   We train the student with mean-square loss using full-batch gradient descent for 1000 epochs with a learning rate of 0.01 and momentum 0.9[4]. We examine the effect of regularization by adding weight decay with parameter 0.05, and the effect of ensembling by averaging over 10 initialization seeds for the weights. All results are averaged over these 10 runs.

## 1.3   Test loss phase space

In both models, the key quantity of interest is the *test loss*, defined as the mean-square loss evaluated on fresh samples $\boldsymbol{x} \sim \mathcal{N}(0, 1)$: $\mathcal{L}_g = \mathbb{E}_{\boldsymbol{x}}\left[(f(\boldsymbol{x}) - f^{\star}(\boldsymbol{x}))^2\right]$.

In the RF model, this quantity was first derived rigorously in [11], in the high-dimensional limit where $N, P, D$ are sent to infinity with their ratios finite. More recently, a different approach based on the Replica Method from Statistic Physics was proposed in [37]; we use this method to compute the analytical phase space. As for the NN model, which operates at finite size $D = 196$, the test loss is computed over a test set of $10^4$ examples.

In Fig. 3, we plot the test loss as a function of two intensive ratios of interest: the number of parameters per dimension $P/D$ and the number of training examples per dimension $N/D$. In the left panel, at high SNR, we observe an overfitting line at $N = P$, yielding a parameter-wise and sample-wise double descent. However when the SNR becomes smaller than unity (middle panel), the sample-wise profile undergoes triple descent, with a second overfitting line appearing at $N = D$. A qualitatively identical situation is shown for the NN model in the right panel[5].

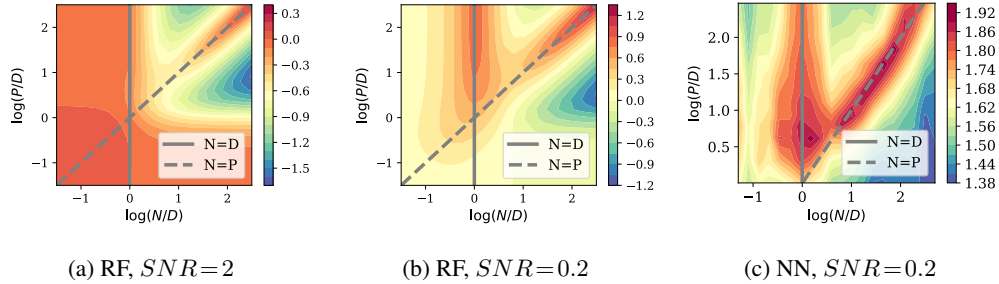

(a) RF, $SNR=2$      (b) RF, $SNR=0.2$      (c) NN, $SNR=0.2$

Figure 3: Logarithmic plot of the test loss in the $(P, N)$ phase space. **(a)**: RF model with SNR $= 2$, $\gamma = 10^{-1}$. **(b)**: RF model with SNR $= 0.2$, $\gamma = 10^{-1}$. The solid arrows emphasize the sample-wise profile, and the dashed lines emphasize the parameter-wise profile. **(c)**: NN model. In all cases, $\sigma = \mathrm{Tanh}$. Analogous results for different activation functions and values of the SNR are shown in Sec. A of the SM.

**The case of structured data** The case of structured datasets such as CIFAR10 is discussed in Sec. C of the SM. The main differences are (i) the presence of multiple linear peaks at $N < D$ due to the complex covariance structure of the data, as observed in [19, 35], and (ii) the fact that the nonlinear peak is located sligthly above the line $N = P$ since the data is easier to fit, as observed in [18].

## 2 Theory for the RF model

The qualitative similarity between the central and right panels of Fig. 3 indicates that a full understanding can be gained by a theoretical analysis of the RF model, which we present in this section.

### 2.1 High-dimensional setup

As is usual for the study of RF models, we consider the following high-dimensional limit:

$$N, D, P \to \infty, \quad \frac{D}{P} = \psi = \mathcal{O}(1), \quad \frac{D}{N} = \phi = \mathcal{O}(1) \tag{4}$$

Then the key quantities governing the behavior of the system are related to the properties of the nonlinearity around the origin:

$$\eta = \int dz \frac{e^{-z^2/2}}{\sqrt{2\pi}} \sigma^2(z), \quad \zeta = \left[ \int dz \frac{e^{-z^2/2}}{\sqrt{2\pi}} \sigma'(z) \right]^2 \quad \text{and} \quad r = \frac{\zeta}{\eta} \tag{5}$$

As explained in [41], the Gaussian Equivalence Theorem [11, 42, 41] which applies in this high dimensional setting establishes an equivalence to a *Gaussian covariate model* where the nonlinear activation function is replaced by a linear term and a nonlinear term acting as noise:

$$\boldsymbol{Z} = \sigma \left( \frac{\boldsymbol{X}\boldsymbol{\Theta}^\top}{\sqrt{D}} \right) \to \sqrt{\zeta} \frac{\boldsymbol{X}\boldsymbol{\Theta}^\top}{\sqrt{D}} + \sqrt{\eta - \zeta} \boldsymbol{W}, \quad \boldsymbol{W} \sim \mathcal{N}(0, 1) \tag{6}$$

Of prime importance is the *degree of linearity* $r = \zeta/\eta \in [0, 1]$, which indicates the relative magnitudes of the linear and the nonlinear terms[6].

## 2.2 Spectral analysis

As expressed by Eq. 3, RF regression is equivalent to linear regression on a structured dataset $\boldsymbol{Z} \in \mathbb{R}^{N \times P}$, which is projected from the original i.i.d dataset $\boldsymbol{X} \in \mathbb{R}^{N \times D}$. In [6], it was shown that the peak which occurs in unregularized linear regression on i.i.d. data is linked to vanishingly small (but non-zero) eigenvalues in the covariance of the input data. Indeed, the norm of the interpolator needs to become very large to fit small eigenvalues according to Eq.3, yielding high variance.

Following this line, we examine the eigenspectrum of $\boldsymbol{\Sigma} = \frac{1}{N} \boldsymbol{Z}^{\top} \boldsymbol{Z}$, which was derived in a series of recent papers. The spectral density $\rho(\lambda)$ can be obtained from the resolvent $G(z)$ [38, 43, 44, 45]:

$$\rho(\lambda) = \frac{1}{\pi} \lim_{\epsilon \to 0^+} \mathrm{Im} G(\lambda - i\epsilon), \qquad G(z) = \frac{\psi}{z} A\left(\frac{1}{z\psi}\right) + \frac{1 - \psi}{z}$$

$$A(t) = 1 + (\eta - \zeta) t A_\phi(t) A_\psi(t) + \frac{A_\phi(t) A_\psi(t) t \zeta}{1 - A_\phi(t) A_\psi(t) t \zeta} \tag{7}$$

where $A_\phi(t) = 1 + (A(t) - 1)\phi$ and $A_\psi(t) = 1 + (A(t) - 1)\psi$. We solve the implicit equation for $A(t)$ numerically, see for example Eq. 11 of [38].

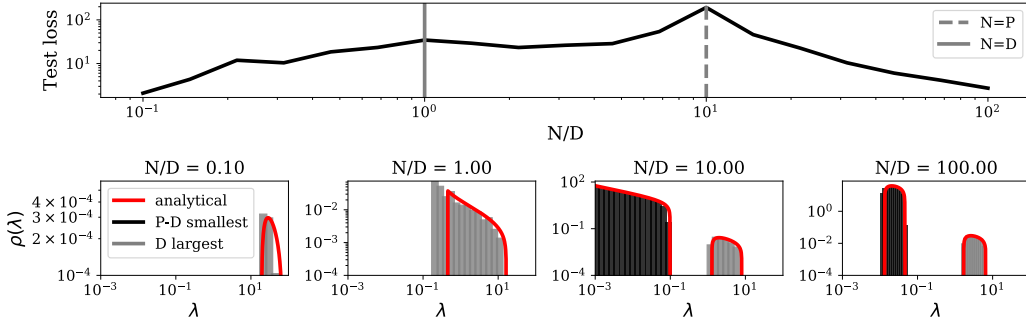

Figure 4: Empirical eigenspectrum of the covariance of the projected features $\boldsymbol{\Sigma} = \frac{1}{N} \boldsymbol{Z}^{\top} \boldsymbol{Z}$ at various values of $N/D$, with the corresponding test loss curve shown above. Analytics match the numerics even at $D = 100$. We color the top $D$ eigenvalues in gray, which allows to separate the linear and nonlinear components at $N > D$. We set $\sigma = \mathrm{Tanh}$, $P/D = 10$, $\mathrm{SNR} = 0.2$, $\gamma = 10^{-5}$.

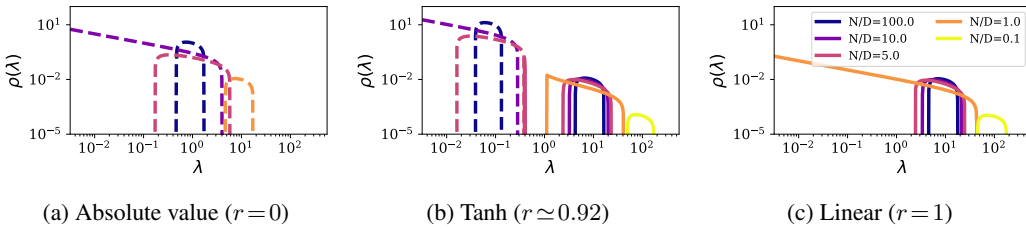

(a) Absolute value ($r = 0$)     (b) Tanh ($r \simeq 0.92$)     (c) Linear ($r = 1$)

Figure 5: Analytical eigenspectrum of $\boldsymbol{\Sigma}$ for $\eta = 1$, $P/D = 10$, and $\zeta = 0, 0.92, 1$ (**a,b,c**). We distinguish linear and nonlinear components by using respectively solid and dashed lines. (**a**) (purely nonlinear): the spectral gap vanishes at $N = P$ (i.e. $N/D = 10$). (**c**) (purely linear): the spectral gap vanishes at $N = D$. (**b**) (intermediate): the spectral gap of the nonlinear component vanishes at $N = P$, but the gap of the linear component does not vanish at $N = D$.

In the bottom row of Fig. 4 (see also middle panel of Fig. 5), we show the numerical spectrum obtained for various values of $N/D$ with $\sigma = \mathrm{Tanh}$, and we superimpose the analytical prediction obtained from Eq. 7. At $N > D$, the spectrum separates into two components: one with $D$ large eigenvalues, and the other with $P - D$ smaller eigenvalues. The spectral gap (distance of the left edge of the spectrum to zero) closes at $N = P$, causing the nonlinear peak [46], but remains finite at $N = D$.

Fig. 5 shows the effect of varying $r$ on the spectrum. We can interpret the results from Eq. 6:

- **"Purely nonlinear"** ($r=0$): this is the case of even activation functions such as $x \mapsto |x|$, which verify $\zeta = 0$ according to Eq. 5. The spectrum of $\mathbf{\Sigma}_{nl} = \frac{1}{N}\mathbf{W}^\top\mathbf{W}$ follows a Marcenko-Pastur distribution of parameter $c = P/N$, concentrating around $\lambda = 1$ at $N/D \to \infty$. The spectral gap closes at $N = P$.

- **"Purely linear"** ($r=1$): this is the maximal value for $r$, and is achieved only for linear networks. The spectrum of $\mathbf{\Sigma}_l = \frac{1}{ND}(\mathbf{X}\mathbf{\Theta}^\top)^\top\mathbf{X}\mathbf{\Theta}^\top$ follows a product Wishart distribution [47, 48], concentrating around $\lambda = P/D = 10$ at $N/D \to \infty$. The spectral gap closes at $N = D$.

- **Intermediate** ($0 < r < 1$): this case encompasses all commonly used activation functions such as ReLU and Tanh. We recognize the linear and nonlinear components, which behave almost independently (they are simply shifted to the left by a factor of $r$ and $1-r$ respectively), except at $N = D$ where they interact nontrivially, leading to implicit regularization (see below).

**The linear peak is implicitly regularized** As stated previously, one can expect to observe over-fitting peaks when $\mathbf{\Sigma}$ is badly conditioned, i.e. its when its spectral gap vanishes. This is indeed observed in the purely linear setup at $N = D$, and in the purely nonlinear setup at $N = P$. However, in the everyday case where $0 < r < 1$, the spectral gap only vanishes at $N = P$, and not at $N = D$. The reason for this is that a vanishing gap is symptomatic of a random matrix reaching its maximal rank. Since $\mathrm{rk}(\mathbf{\Sigma}_{nl}) = \min(N, P)$ and $\mathrm{rk}(\mathbf{\Sigma}_l) = \min(N, P, D)$, we have $\mathrm{rk}(\mathbf{\Sigma}_{nl}) \geq \mathrm{rk}(\mathbf{\Sigma}_l)$ at $P > D$. Therefore, the rank of $\mathbf{\Sigma}$ is imposed by the nonlinear component, which only reaches its maximal rank at $N = P$. At $N = D$, the nonlinear component acts as an *implicit regularization*, by compensating the small eigenvalues of the linear component. This causes the linear peak to be implicitly regularized be the presence of the nonlinearity.

**What is the linear peak caused by?** At $0 < r < 1$, the spectral gap vanishes at $N = P$, causing the norm of the estimator $\|\mathbf{a}\|$ to peak, but it does not vanish at $N = D$ due to the implicit regularization; in fact, the lowest eigenvalue of the full spectrum does not even reach a local minimum at $N = D$. Nonetheless, a soft *linear peak* remains as a vestige of what happens at $r = 1$. What is this peak caused by? A closer look at the spectrum of Fig. 5.b clarifies this question. Although the left edge of the full spectrum is not minimal at $N = D$, the left edge of the *linear component*, in solid lines, reaches a minimum at $N = D$. This causes a peak in $\|\mathbf{\Theta}\mathbf{a}\|$, the norm of the "linearized network", as shown in Sec. B of the SM. This, in turn, entails a different kind of overfitting as we explain in the next section.

### 2.3 Bias-variance decomposition

The previous spectral analysis suggests that both peaks are related to some kind of overfitting. To address this issue, we make use of the bias-variance decomposition presented in [20]. The test loss is broken down into four contributions: a bias term and three variance terms stemming from the randomness of (i) the random feature vectors $\mathbf{\Theta}$ (which plays the role of initialization variance in realistic networks), (ii) the noise $\boldsymbol{\epsilon}$ corrupting the labels of the training set (noise variance) and (iii) the inputs $\mathbf{X}$ (sampling variance). We defer to [20] for further details.

**Only the nonlinear peak is affected by initialization variance** In Fig. 6.a, we show such a decomposition. As observed in [20], the nonlinear peak is caused by an interplay between initialization and noise variance. This peak appears starkly at $N = P$ in the high noise setup, where noise variance dominates the test loss, but also in the noiseless setup (Fig. 6.b), where the residual initialization variance dominates: nonlinear networks can overfit even in absence of noise.

**The linear peak vanishes in the noiseless setup** In stark contrast, the linear peak which appears clearly at $N = D$ in Fig. 6.a is caused solely by a peak in noise variance, in agreement with [6]. Therefore, it vanishes in the noiseless setup of Fig. 6.b. This is expected, as for linear networks the solution to the minimization problem is independent of the initialization of the weights.

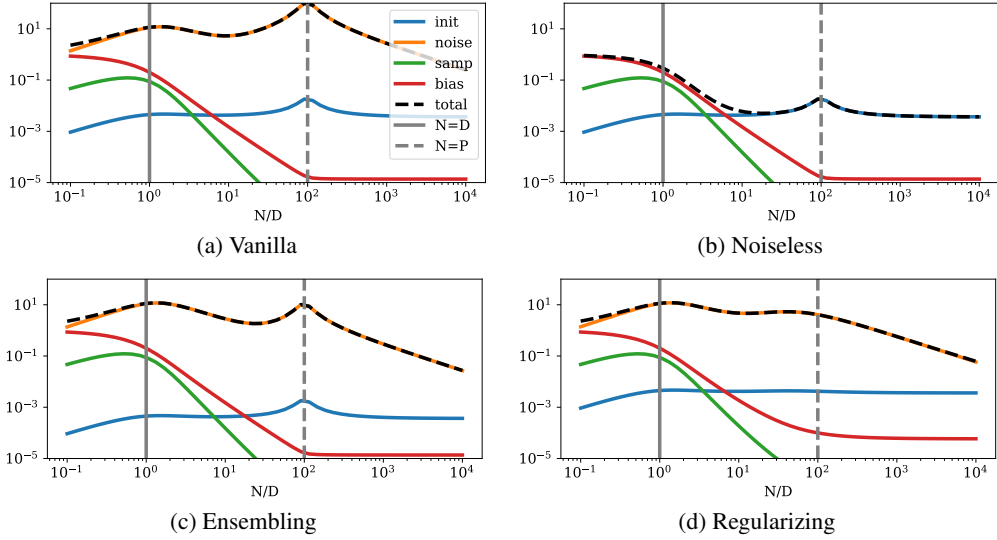

Figure 6: Bias-variance decompostion of the test loss in the RF model for $\sigma = \text{ReLU}$ and $P/D = 100$. Regularizing (increasing $\gamma$) and ensembling (increasing the number $K$ of initialization seeds we average over) mitigates the nonlinear peak but does not affect the linear peak. **(a)** $K=1, \gamma = 10^{-5}, SNR = 0.2$. **(b)** Same but $SNR = \infty$. **(c)** Same but $K = 10$. **(d)** Same but $\gamma = 10^{-3}$.

# 3 Phenomenology of triple descent

## 3.1 The nonlinearity determines the relative height of the peaks

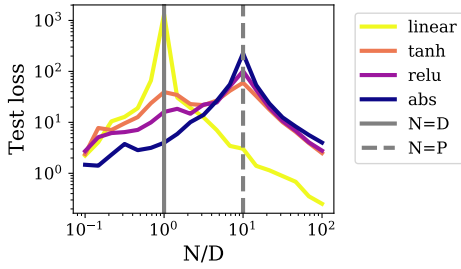

Figure 7: Numerical test loss of RF models at finite size ($D = 100$), averaged over 10 runs. We set $P/D = 10$, SNR $= 0.2$ and $\gamma = 10^{-3}$.

In Fig. 7, we consider RF models with four different activation functions: absolute value ($r = 0$), ReLU ($r = 0.5$), Tanh ($r \sim 0.92$) and linear ($r = 1$). We see that increasing the degree of nonlinearity strengthens the nonlinear peak (by increasing initialization variance) and weakens the linear peak (by increasing the implicit regularization). In Sec. A of the SM, we present additional results where the degree of linearity $r$ is varied systematically in the RF model, and show that replacing Tanh by ReLU in the NN setup produces a similar effect. Note that the behavior changes abruptly near $r = 1$, marking the transition to the linear regime.

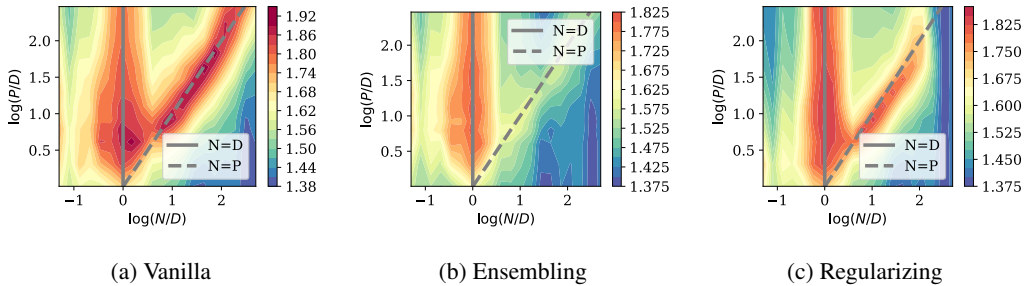

(a) Vanilla     (b) Ensembling     (c) Regularizing

Figure 8: Test loss phase space for the NN model with $\sigma = \text{Tanh}$. Weight decay with parameter $\gamma$ and ensembling over $K$ seeds weakens the nonlinear peak but leaves the linear peak untouched. **(a)** $K=1, \gamma=0, SNR=0.2$. **(b)** Same but $K = 10$. **(c)** Same but $\gamma = 0.05$.

### 3.2 Ensembling and regularization only affects the nonlinear peak

It is a well-known fact that regularization [19] and ensembling [9, 20, 49] can mitigate the nonlinear peak. This is shown in panel (c) and (d) of Fig. 6 for the RF model, where ensembling is performed by averaging the predictions of 10 RF models with independently sampled random feature vectors. However, we see that these procedures only weakly affect the linear peak. This can be understood by the fact that the linear peak is already implicitly regularized by the nonlinearity for $r < 1$, as explained in Sec. 2.

In the NN model, we perform a similar experiment by using weight decay as a proxy for the regularization procedure, see Fig. 8. Similarly as in the RF model, both ensembling and regularizing attenuates the nonlinear peak much more than the linear peak.

### 3.3 The nonlinear peak forms later during training

To study the evolution of the phase space during training dynamics, we focus on the NN model (there are no dynamics involved in the RF model we considered, where the second layer weights were learnt via ridge regression). In Fig. 9, we see that the linear peak appears early during training and maintains throughout, whereas the nonlinear peak only forms at late times. This can be understood qualitatively as follows [6]: for linear regression the time required to learn a mode of eigenvalue $\lambda$ in the covariance matrix is proportional to $\lambda^{-1}$. Since the nonlinear peak is due to vanishingly small eigenvalues, which is not the case of the linear peak, the nonlinear peak takes more time to form completely.

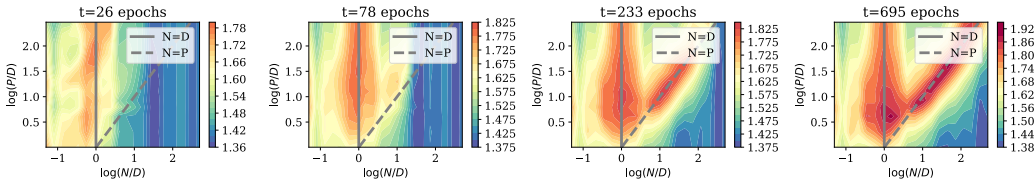

Figure 9: Test loss phase space for the NN model with $\sigma = \text{Tanh}$, plotted at various times during training. The linear peak grows first, followed by the nonlinear peak.

## 4 Conclusion

One of the key challenges in solving tasks with network-like architectures lies in choosing an appropriate number of parameters $P$ given the properties of the training dataset, namely its size $N$ and dimension $D$. By elucidating the structure of the $(P, N)$ phase space, its dependency on $D$, and distinguishing the two different types of overfitting which it can exhibit, we believe our results can be of interest to practitioners.

Our results leave room for several interesting follow-up questions, among which the impact of (1) various architectural choices, (2) the optimization algorithm, and (3) the structure of the dataset. For future work, we will consider extensions along those lines with particular attention to the structure of the dataset. We believe it will provide a deeper insight into data-model matching.

**Acknowledgements**   We thank Federica Gerace, Armand Joulin, Florent Krzakala, Bruno Loureiro, Franco Pellegrini, Maria Refinetti, Matthieu Wyart and Lenka Zdeborova for insightful discussions. GB acknowledges funding from the French government under management of Agence Nationale de la Recherche as part of the "Investissements d'avenir" program, reference ANR-19-P3IA-0001 (PRAIRIE 3IA Institute) and from the Simons Foundation collaboration Cracking the Glass Problem (No. 454935 to G. Biroli).

**Broader Impact**   Due to the theoretical nature of this paper, a Broader Impact discussion is not easily applicable. However, given the tight interaction of data & model and their impact on overfitting regimes, we believe that our findings and this line of research, in general, may potentially impact how practitioners deal with data.

## Footnotes

[1] Also called the *jamming* peak due to similarities with a well-studied phenomenon in the Statistical Physics literature [14, 15, 16, 17, 18].

[2]The name "triple descent" refers to the presence of two peaks instead of just one in the famous "double descent" curve, but in most cases the test error does not actually descend before the first peak.

[3]This model, shown to undergo double descent in [11], has become a cornerstone to study the so-called lazy learning regime of neural networks where the weights stay close to their initial value [39]: assuming $f_{\theta_0} = 0$, we have $f_{\theta}(\mathbf{x}) \approx \nabla_{\theta} f_{\theta}(\mathbf{x})|_{\theta=\theta_0} \cdot (\theta - \theta_0)$ [40]. In other words, lazy learning amounts to a linear fitting problem with a random feature vector $\nabla_{\theta} f_{\theta}(\mathbf{x})|_{\theta=\theta_0}$.

[4]We use full batch gradient descent with small learning rate to reduce the noise coming from the optimization as much as possible. After 1000 epochs, all observables appear to have converged.

[5]Note that for NNs, we necessarily have $P/D > 1$.

[6]Note from Eq. 5 that for non-homogeneous functions such as $\mathrm{Tanh}$, $r$ also depends on the variance of the inputs and fixed weights, both set to unity here: intuitively, smaller variance will yield smaller preactivations which will lie in the linear region of the $\mathrm{Tanh}$, increasing the effective value of $r$.

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
