[Supplementary Material 1]

## A  Effect of signal-to-noise ratio and nonlinearity

### A.1  RF model

In the RF model, varying $r$ can easily be achieved analytically and yields interesting results, as shown in Fig. 10[6].

In the top panel, we see that the parameter-wise profile exhibits double descent for all degrees of linearity $r$ and signal-to-noise ratio SNR, except in the linear case $r = 1$ which is monotonously deceasing. Increasing the degree of nonlinearity (decreasing $r$) and the noise (decreasing the SNR) simply makes the nonlinear peak stronger.

In the bottom panel, we see that the sample-wise profile is more complex. In the linear case $r = 1$, only the linear peak appears (except in the noiseless case). In the nonlinear case $r < 1$, the nonlinear peak appears is always visible; as for the linear peak, it is regularized away, except in the strong noise regime SNR $> 1$ when the degree of nonlinearity is small ($r > 0.8$), where we observe the triple descent.

Notice that both in the parameter-wise and sample-wise profiles, the test loss profiles change smoothly with $r$, except near $r = 1$ where the behavior abruptly changes, particularly at low SNR.

Figure 10: Analytical parameter-wise (**top**, $N/D = 10$) and sample-wise (**bottom**, $P/D = 10$) test loss profiles of the RF model. **Left**: noiseless case, SNR $= \infty$. **Center**: low noise, SNR $= 2$. **Right**: high noise, SNR $= 0.2$. We set $\gamma = 10^{-1}$.

One can also mimick these results numerically by considering, as in [30], the following family of piecewise linear functions:

$$\sigma_\alpha(x) = \frac{[x]_+ + \alpha[-x]_+ - \frac{1+\alpha}{\sqrt{2\pi}}}{\sqrt{\frac{1}{2}\left(1 + \alpha^2\right) - \frac{1}{2\pi}(1+\alpha)^2}},  \tag{8}$$

for which

$$r_\alpha = \frac{(1-\alpha)^2}{2\left(1+\alpha^2\right) - \frac{2}{\pi}(1+\alpha)^2}.  \tag{9}$$

Here, $\alpha$ parametrizes the ratio of the slope of the negative part to the positive part and allows to adjust the value of $r$ continuously. $\alpha = -1$ ($r = 1$) will correspond to a (shifted) absolute value, $\alpha = 1$ ($r = 0$) will correspond to a linear function, $\alpha = 0$ will correspond to a (shifted) ReLU. In Fig. 11, we show the effect of sweeping $\alpha$ uniformly from 1 to -1 (which causes $r$ to range from 0 to 1). As expected, we see the linear peak become stronger and the nonlinear peak become weaker.

(a) Nonlinearities used

(b) Corresponding test loss

Figure 11: Moving from a purely nonlinear function to a purely linear function (dark to light colors) strengthens the linear peak and weakens the nonlinear peak.

## A.2 NN model

We show in the top row of Fig. 12 the effect of varying the SNR on the $(P, N)$ phase space for $\sigma = \mathrm{Tanh}$ in the NN model. Just like in the RF model, triple descent only appears at $\mathrm{SNR} < 1$ (right panel).

In the bottom row of the same figure, we show the effect of replacing Tanh ($r \sim 0.92$) by ReLU ($r = 0.5$). In the low SNR setup, we still distinguish the two peaks of triple descent, but the linear peak is much weaker, as expected from the stronger degree of nonlinearity.

Notice that in the intermediate signal-to-noise scenario, $1 < \mathrm{SNR} < \infty$, results are different from the RF model where we only observed the nonlinear peak. For Tanh, we observe only the linear peak, whereas for ReLU, we observe something intermediate between the linear peak and the nonlinear peak.

## B   Origin of the linear peak

In this section, we follow the lines of [28], where the test loss is decomposed in the following way (Eq. D.6):

$$\mathcal{L}_g = \rho + Q - 2M \tag{10}$$

$$\rho = \frac{1}{D}\|\boldsymbol{\beta}\|^2, \quad M = \frac{\sqrt{\zeta}}{D}\boldsymbol{b}\cdot\boldsymbol{\beta}, \quad Q = \frac{\zeta}{D}\|\boldsymbol{b}\|^2 + \frac{\eta - \zeta}{P}\|\boldsymbol{a}\|^2, \quad \boldsymbol{b} = \boldsymbol{\Theta}\boldsymbol{a} \tag{11}$$

As before, $\boldsymbol{\beta}$ denotes the linear teacher vector and $\boldsymbol{\Theta}, \boldsymbol{a}$ respectively denote the (fixed) first and (learnt) second layer of the student. This insightful expression shows that the loss only depends on the norm of the second layer $\|\boldsymbol{a}\|$, the norm of the linearized network $\|\boldsymbol{b}\|$, and its overlap with the teacher $\boldsymbol{b}\cdot\boldsymbol{\beta}$.

We plot these three terms in Fig. 13, focusing on the triple descent scenario $\mathrm{SNR} < 1$. In the left panel, we see that the overlap of the student with the teacher is monotically increasing, and reaches its maximal value at a certain point which increases from $D$ to $P$ as we decrease $r$ from 1 to 0. In the central panel, we see that $\|\boldsymbol{a}\|$ peaks at $N = P$, causing the nonlinear peak as expected, but nothing special happens at $N = D$ (except for $r = 1$). However, in the right panel, we see that the norm of the linearized network peaks at $N = D$, where we know from the spectral analysis that the gap of the linear part of the spectrum is minimal. This is the origin of the linear peak.

(a) Tanh, $SNR = \infty$       (b) Tanh, $SNR = 2$       (c) Tanh, $SNR = 0.2$

(d) ReLU, $SNR = \infty$       (e) ReLU, $SNR = 2$       (f) ReLU, $SNR = 0.2$

Figure 12: Logarithmic plot of the test loss in the phase space defined by number of parameters. **Left**: Single descent at low SNR. **Center**: Double descent at intermediate SNR. **Right**: Triple descent at low SNR.

Figure 13: Terms entering Eq. 11, plotted at $SNR = 0.2$, $\gamma = 10^{-1}$.

## C    Structured datasets

In this section, we examine how our results are affected by considering the realistic case of correlated data. To do so, we replace the Gaussian i.i.d. data by MNIST data, downsampled to $10 \times 10$ images for the RF model ($D = 100$) and $14 \times 14$ images for the NN model ($D = 196$).

### C.1    RF model

We refer to the results in Fig 14. Interestingly, the triple descent profile is weakly affected by the correlated structure of this realistic dataset. However, the spectral properties of $\boldsymbol{\Sigma} = \frac{1}{N} \boldsymbol{Z}^\top \boldsymbol{Z}$ are changed in an interesting manner: the two parts of the spectrum are now contiguous, there is no gap between the linear part and the nonlinear part.

(a) Random data

(b) MNIST

Figure 14: Spectrum of the covariance of the projected features $\boldsymbol{\Sigma} = \frac{1}{N}\boldsymbol{Z}^\top\boldsymbol{Z}$ at various values of $N/D$, with the corresponding loss curve shown above. We set $\sigma = \text{Tanh}, \gamma = 10^{-5}$.

## C.2  NN model

As shown in the top row of Fig. 15, the NN model is qualitatively different on the structured dataset: the two peaks at $N = D$ and $N = P$ are not well separated at $\text{SNR} < 1$ anymore. The single peak which appears is somewhat intermediate between the $N = D$ and $N = P$. However, by considering the time evolution in the bottom row of the same figure, we see that this peak shifts across the phase space during training, just like in the case of random data (Fig. 9).

At early times, it is located along a line of constant $N$, which makes it akin to a linear peak. At late times, it is rather reminiscent of a nonlinear peak, though it does not seem to be located at $P \sim N$ as before, but rather at $N \sim P^\alpha$ with $\alpha < 1$. This sublinear scaling is a consequence of the fact that structured data is easier to memorize than random data [17], and may blur the distinction between the two peaks.

Interestingly, at early times, the peak does not occur at $N = D$ as expected, but rather at $N = D_{\text{eff}} \sim D/10 \sim 20$. We hypothesize that $D_{\text{eff}}$ may be related to the intrinsic dimension of the input data [39, 40, 41]. Although the linear peak still occurs at $N = D$ for MNIST data in the RF model, in the NN setup feature learning occurs. When the dataset is highly correlated like MNIST, feature learning compresses the dataset down to a more compact representation, likely causing the $N = D$ peak to shift to lower values. A study of this crucial question is deferred to future work.

(a) MNIST, $SNR = \infty$

(b) MNIST, $SNR = 2$

(c) MNIST, $SNR = 0.2$

t=37 epochs

t=162 epochs

t=695 epochs

(d) Dynamics on MNIST at $SNR = 0.2$

Figure 15: Test loss phase space on MNIST with $\sigma = \mathrm{ReLU}$. **Top**: After 1000 epochs, for various values of the SNR. **Bottom**: at three different times during training in the low SNR case.

## Footnotes

[6]We focus here on the practically relevant setup $N/D \gg 1$. Note from the $(P, N)$ phase-space that things can be more complex at $N/D \lesssim 1$).


[Supplementary Material 2 · spectra_random_relu.pdf]



Test loss vs N/D with legend: N=P (dashed), N=D (solid).

Bottom row of spectral density plots $\rho(\lambda)$ vs $\lambda$ for: N/D = 0.10, N/D = 0.32, N/D = 1.00, N/D = 3.16, N/D = 10.00, N/D = 31.62, N/D = 100.00

[Supplementary Material 3]



$X \in \mathbb{R}^{N \times D}$

$Z = \sigma(\frac{1}{\sqrt{D}} X \Theta^T) \in \mathbb{R}^{N \times P}$

$y = a Z^\top \in \mathbb{R}^N$

$a \in \mathbb{R}^P$

$\Theta \in \mathbb{R}^{P \times D}$



[Supplementary Material 4]



Test loss vs N/D with legend: N=P (dashed), N=D (solid)

Bottom row panels ($\rho(\lambda)$ vs $\lambda$): N/D = 0.10, N/D = 0.32, N/D = 1.00, N/D = 3.16, N/D = 10.00, N/D = 31.62, N/D = 100.00

[Supplementary Material 5 · perfs_MNIST_r.pdf]



| | |
|---|---|
| ▬ | -1.0 |
| ▬ | -0.8 |
| ▬ | -0.6 |
| ▬ | -0.4 |
| ▬ | -0.2 |
| ▬ | 0.0 |
| ▬ | 0.2 |
| ▬ | 0.4 |
| ▬ | 0.6 |
| ▬ | 0.8 |
| ▬ | 1.0 |
| ▬ | N=D |
| ┄ | N=P |