[Reviews · NeurIPS 2020]

Review 1

Summary and Contributions: Update: I'm grateful for the authors' time in responding to the reviewers' questions. However, I was disappointed they did not clarify whether whether any new theoretical results are proved in the paper. Specifically, whether their claim that "the linear peak at N=D is solely due to overfitting the noise in the labels" has a precise, mathematical statement and proof. Addressing this in the paper's introduction will help readers understand the novelty and impact. ___________________________________________________________________________ Random feature regression in the high-dimensional limit has been a useful model for double descent behavior, where the test loss decreases, increases, and then decreases again as a function of the number of hidden features for a fixed dataset. The authors discuss the existence of a triple descent phenomenon in the test loss as a function of the dataset size. Specifically, they find a second point of nonmonotonicity when the dataset size and dimensionality are equal. Furthermore, they analyze the test loss using a bias-variance decomposition developed in previous work.

Strengths: The author's discussion of the descent when the dataset size and dimensionality are equal is extensive and has not be highlighted before. They give two interesting perspectives to explain this behavior, namely an algebraic one based on the rank and a bias-variance perspective. The paper also has many excellent figures that help the presentation and aid in the understanding of their results.

Weaknesses: The paper's primary weakness is that it seems to overload the definition of double descent. Traditionally, the test loss is viewed as a function of the model complexity for a fixed training dataset. This is an advantage of the RF model over previous kernel regression models of double descent, where the model complexity is tied to the training data. It turns out that in this high-dimensional random features setting, only the ratios between the dimensions are relevant, so it is tempting to think that viewing the test loss as a function of width or dataset size is symmetric. However, changing the dataset size, changes its relative size wrt. the data dimension. Fundamentally, this is the reason for the behavior. I'm sure the authors understand this. I just feel it makes the result's relationship with double descent quite different and that it is likely to confuse people. A second question is whether any new theoretical results, over those of Mei and Montanari, 2019, and d'Ascoli et al., 2020, are required? It seems like the model is the same and the bias-variance decomposition is the same as d'Ascoli. Could the authors clarify this?

Correctness: The results all look correct.

Clarity: I found the paper well written, easy to follow, and an enjoyable read. In particular, the figures are excellent are really benefit the paper. Some of the legends are quite short though. With more room perhaps the authors should consider expanding them.

Relation to Prior Work: The paper is very well referenced and situates itself properly in the field. As I mentioned above, some more details on whether any new theoretical results are required above Mei and Montanari, 2019, and d'Ascoli et al., 2020, would be good. The paper of Liang et al., "on the multiple descent of minimum-norm interpolants and restricted lower isometry of kernels" is relevant. The authors should also discuss differences with Adlam and Pennington, 2020, which also finds a triple descent phenomenon.

Reproducibility: Yes

Additional Feedback: Line 163: "at" should be "as." Line 224: There can be training dynamics associated with the RF model. Specifically, it is related to a NN with random first-layer weights, but second-layer weights initialized at 0 and trained via gradient descent.


Review 2

Summary and Contributions: This paper takes a closer look at the "double descent" and related phenomena, where models "overfit" in some narrow regime of parameters, but behave well outside this regime. This work considers the test risk as a function of: input-dimension (D), model dimension (P), and number of train points (N). It considers stylized models (random-features in theory, and 2-layer networks in practice) which have been studied in previous works, and are reasonable models for studying double-descent. There are two main contributions: 1. By looking at the landscape as a function of both D and P, they find two distinct kinds of "overfitting" -- one determined by P and one by D. This paper is, as far as I know, the first to point out these two distinct effects. 2. They theoretically analyze these two effects, and show how they depend on problem parameters including the type of non-linearity. (Building on recent works which related the test risk to the spectrum of certain random matrices). These are significant conceptual contributions which clarify the landscape of overfitting. They only apply to restricted models (random features and 2 layer nets), but it is a promising first step to studying the phenomenon more broadly.

Strengths: The main strengths of this work are its conceptual novelty: pointing out two distinct kinds of overfitting which have been conflated in the past. An additional strength is the mathematical analysis: the expressions for test risk have existed in past work, but were not studied carefully enough to decouple the two effects presented in this work. It is significant that the mathematical analysis is able to yield insight into the empirical effects. Broader relevance: This paper continues the line of work studying overparameterized models and their pathologies (double-descent, etc.). This topic has had a resurgence of interest recently, and should be of interest to the NeurIPS community.

Weaknesses: One weakness of this work is the theory and experiments are in toy settings: random features and 2-layer networks on simple ground-truth distributions (linear, or random-relu-teacher). This is necessary for analytical tractability, but because there are no experiments on real data, it is unclear whether the insights obtained are relevant for neural networks in practice. It is of course independently interesting to study the dynamics of these toy models, but it would be much more significant if these effects were also demonstrated or investigated on real data. For example, are the two kinds of overfitting empirically present in say convnets on CIFAR (or even 2-layer nets on MNIST)? If so, that would strengthen this paper's impact significantly. If not, why not? -- what separates these more realistic settings from the ones in this paper?

Correctness: There are no major technical issues as far as I am aware. Some minor issues: Section 3.2 only tests one fixed value of regularization. If regularization is studied, it makes more sense to optimize over the regularization parameter, and consider only the optimal choice for each setting of (N, D, P). Section 3.3 says "Since the nonlinear peak is due to vanishingly small eigenvalues, which is not the case of the linear peak,..." Aren't both peaks due to small eigenvalues in Z^TZ?

Clarity: The clarity of the paper could be improved. While the gist of the paper is clear, many of the most interesting details are not presented clearly. Figure 5 is the primary figure which supports the theoretical part of this paper. More effort should be put into making this figure clear. For example, overlaying all of the plots for varying N/D ratios creates visual confusion, and obscures the "gaps" -- which is the crucial part. I suggest separate plots for each N/D, so that the reader can focus on one setting at a time. The figures should also be annotated to highlight the interesting features -- the "linear" and "nonlinear" contribution to the spectrum, and the gaps between them. The text discusses how this gap depends on N/D, although this is unclear from the current figures. More comments: - Line 176-177 ("the nonlinear component acts as an implicit regularization…") sounds interesting and should be explained more. More generally, the section "Analysis of gaps" should explain more why studying the gap is interestering/important. - The "3D" plots may be fine once, but I suggest moving to flat 2d scalar-field plots. (3d obscures some of the data and makes comparison across graphs hard, since the z-axis is scaled differently each time). - The top figure of Figure 4 is not referenced in the text, and it's unclear what this figure is adding. - Section 2.3 can be removed, or moved to the appendix. The bias-variance has been done in many prior papers by now (including the ones cited). This section does not appear to contain new insights beyond prior work, which already discussed ensembling and regularization. Moreover, the text is too short to be clarifying, and Figure 6 is too dense to parse without discussion (there are 7 overlapping lines per plot, and 4 plots). If there is some message in this section that the authors want to convey, it should be conveyed much more clearly. - Section 3.1, on the effect of nonlinearity, is interesting and deserves more discussion.

Relation to Prior Work: Prior work needs to be discussed more extensively -- several important citations are missing or incorrect. The work [A] was one of the first to explicitly study the test loss landscape as a function of model-size and sample-size independently (as opposed to just via their ratio), which is a key aspect of this work. As far as I am aware, [A] introduced the term "sample-wise non-monotonicity", which is used throughout this paper, but is not cited. The first citation to [25], on line 75, is incorrect. [25] is not about adversarial training. There are several references to earlier work missing from Line 36 ("a similar phenomenon has been well-known for several decades for simpler models"): see [C] and [D] from 1990, 1991. These are even earlier than the cited works (from 1996). Monotonicity of learning algorithms was also studied in [B]. The relation to prior work [12] should be discussed further. It is my understanding that [12] derived the same asymptotics which are studied here, but studied them under a different regime of parameters. The "triple descent" from [18] is discussed only briefly -- it is mentioned that [18] can be considered as "two linear peaks". This seems worthy of more discussion, especially in the context of this paper: If two linear peaks are possible, then are two non-linear peaks possible as well? In general, this suggests that the picture can be more complex than the "one linear, one nonlinear" story presented in this work. This may be a topic worthy of future work, but some acknowledgement of the complexity of the general setting would clarify even the current work. [A] Deep Double Descent: Nakkiran, Kaplun, Bansal, Yang, Barak, Sutskever. 2019. [B] Minimizers of the Empirical Risk and Risk Monotonicity: Loog, Viering, Mey. 2019. [C] Eigenvalues of covariance matrices: Application to neural-network learning. Le Cun, Kanter, Solla. 1991. [D] Second Order Properties of Error Surfaces: Learning Time and Generalization. Le Cun, Kanter, Solla. 1990.

Reproducibility: Yes

Additional Feedback:


Review 3

Summary and Contributions: The paper shows that when you increase the number of samples used for training a model, the expected test error may exhibit a 'triple descent'. The show that this sample-wise triple descent arises in a random features model depending on the non-linearity, which in turn affects the spectrum of the covariates in the regression. They experimentally confirm that this phenomenon arises in a teacher-student setup.

Strengths: This paper adds to the growing body of literature on double descent in machine learning models and shows that there may also sometimes be a triple descent with the number of samples. The paper uses a random features model to analyze this phenomenon and shows that the occurrence of this depends on the non-linearity used in the random features.

Weaknesses: Post-rebuttal: After reading the author responses and reviewer discussion, I think the paper does make a new conceptual contribution by combining different insights even if the technical novelty is not as strong. I have updated my score and look forward to reading future work on more structured datasets. _____________ Theoretically, the paper does not make a very novel contribution. The main results of the paper rely on the following insights (a) If the smallest eigenvalue of the co-variance matrix decreases with N, this can cause non-monotonicity in the test error -- this has now been studied in various papers including Advani & Saxe (b) The non-linearity can be decomposed into a linear and non-linear term - this was shown in previous work as cited by the authors (c) The spectrum from these two part remain almost independent - this was only shown empirically Moreover, the evidence that this phenomenon may occur outside of the narrow random feature setting is not very strong. The authors do show that the NN model displays this peak, but even in this case the peak is not clear for ReLU as shown in the supplement. There is also no evidence of this occurring for structured data (There are some experiments in the supplement for MNIST but they do not support the hypothesis as strongly) Due to a combination of these two factors, it is not very clear to me what the broader takeaways are from this paper (outside of the possible existence of a triple descent)

Correctness: The theoretical claims for the random features model and the subsequent conclusions are correct. The empirical methodology for the NN model is also correct. Though not central to the main claims, the paper repeatedly mentions that the interpolation threshold is observed when the number of parameters is the same order as the number of samples (eg Line 40-42). The location of the peak can depend on the distribution, architecture and the training algorithm and will not necessarily occur at the order of parameters.

Clarity: There are some areas where the paper could use improvement 1. The plots in the paper are hard to read (Fig 3, 8, 9 and similar figures in the supplement). I would suggest that these be plotted as a 2D figure to improve readability. 2. The results for the NN model are scattered throughout the paper and it would be good to see a summary of what the results are and how much they depend on specific model/data choices. 3. The key takeaways from the paper are not very clear

Relation to Prior Work: The paper has some missing citations to relevant work. [1] should be cited in the introduction when referring to works that show double descent in deep neural networks and in reference to sample-wise double descent. [2] should also be cited in relation to double descent. [1] Nakkiran, P., Kaplun, G., Bansal, Y., Yang, T., Barak, B., & Sutskever, I. (2019). Deep double descent: Where bigger models and more data hurt. arXiv preprint arXiv:1912.02292. Chicago [2] Nakkiran, P. (2019). More data can hurt for linear regression: Sample-wise double descent. arXiv preprint arXiv:1912.07242.

Reproducibility: Yes

Additional Feedback:


Review 4

Summary and Contributions: This paper investigates generalization error of random features regression model as a function of sample size / model complexity, and shows that there can be not one but two distinct error peaks: the "nonlinear" peak when sample size equals model complexity N=P; and the "linear" peak when sample size equals input dimensionality N=D. The paper shows that the same can happen in a neural network model.

Strengths: The paper explores the "double descent" phenomenon that has been focus of much work recently. It clearly elucidates a surprising phenomenon that "double descent" can become "triple descent" with two distinct risk peaks. The peaks are shown to have different properties (e.g. only the nonlinear peak survives in the noiseless regime). I think this is an important contribution a clear accept. Here is the intuitive picture that I formed after reading the paper: imagine a neural network with some hidden layers. It's obvious that when the activation function is linear, the interpolation threshold and the risk peak is at N=D. It's obvious that when the activation function is nonlinear, the interpolation threshold and the risk peak is N=P>D. The paper asks: what happens with the risk peak when the activation function gradually changes from nonlinear to linear? Would the peak slowly move from P to D? The (surprising?) answer is no: the peak at P will go down, the peak at D will go up, and for some time both will co-exist.

Weaknesses: All mathematical development and results here seem to be taken from prior work (so this paper is not math heavy; it's not a weakness but just a remark), but they are well used to explain empirical findings. I do not have any major criticisms but only some more or less minor suggestions to improve the presentation.

Correctness: yes

Clarity: yes

Relation to Prior Work: yes

Reproducibility: Yes

Additional Feedback: This paper investigates generalization error of random features regression model as a function of sample size / model complexity, and shows that there can be not one but two distinct error peaks: the "nonlinear" peak when sample size equals model complexity N=P; and the "linear" peak when sample size equals input dimensionality N=D. The paper shows that the same can happen in a neural network model. The paper explores the "double descent" phenomenon that has been focus of much work recently. It clearly elucidates a surprising phenomenon that "double descent" can become "triple descent" with two distinct risk peaks. The peaks are shown to have different properties (e.g. only the nonlinear peak survives in the noiseless regime). I think this is an important contribution a clear accept. Here is the intuitive picture that I formed after reading the paper: imagine a neural network with some hidden layers. It's obvious that when the activation function is linear, the interpolation threshold and the risk peak is at N=D. It's obvious that when the activation function is nonlinear, the interpolation threshold and the risk peak is N=P>D. The paper asks: what happens with the risk peak when the activation function gradually changes from nonlinear to linear? Would the peak slowly move from P to D? The (surprising?) answer is no: the peak at P will go down, the peak at D will go up, and for some time both will co-exist. All mathematical development and results here seem to be taken from prior work (so this paper is not math heavy; it's not a weakness but just a remark), but they are well used to explain empirical findings. I do not have any major criticisms but only some more or less minor suggestions to improve the presentation. Major comments * none [but please respond at least to the last two bullet points in "Medium comments" during the author response period] Medium comments * Line 35: the literature review for linear regression can be improved: a) Refs [13,14,20,21] are not all "several decades" old. I'd suggest to split them into those that are indeed old [13,14] and then write something like "... and has recently been studied in-depth [20,21]". b) Also, both old/new lists can be extended. For the old: there is Opper 1995 (in The Handbook of Brain Theory and Neural Networks) and there is Duin 1995 http://www.rduin.nl/papers/scia_95.sssize.pdf. There is also Krogh and Hertz 1992 https://iopscience.iop.org/article/10.1088/0305-4470/25/5/020. And Opper 1990: https://iopscience.iop.org/article/10.1088/0305-4470/23/11/012. Maybe cite one paper from each group of authors. Can also cite this recent comment https://www.pnas.org/content/117/20/10625 "A brief prehistory of double descent". c) For the new: consider adding https://arxiv.org/abs/1805.10939 (in press in JMLR). Disclaimer: I am one of the authors. If you think this paper is irrelevant then feel free not to cite it. Two very recent works in the same vein: https://arxiv.org/abs/2006.05800, https://arxiv.org/abs/2006.06386. * Figure 2: as drawn, it looks like P<D, but most subsequent figures (e.g. Fig 4 etc) show examples with P>D. Consider modifying Figure 2 to make P>D. * Figure 2: consider adding another panel for the neural network model described in section 1.2. Line 109 says "with 2 layers" but it's actually not entirely clear if these are 2 hidden layers? or only 1 hidden layer? A figure would make this obvious. * Wouldn't Figure 3 (and later Figs 8/9) be clearer in 2D instead of 3D? You use color coding for the error anyway. The "curves" would of course become simply straight lines which is probably what you did not want; but the color might be clear enough to show non-monotonic error behaviour along these lines. At least to me, 3D plots are not very easy to comprehend. * Good point is raised in lines 128-130 about structured data. The papers I mentioned above in the first bullet point, item (c), deal exactly with this situation of structured data (and non-random true beta vector) and show that in linear regression this can bring out a whole range of new phenomenona (e.g. https://arxiv.org/abs/1805.10939). * Line 162 / Fig 5: why is |x| purely nonlinear (r=0)? This is not very intuitive to me, can you insert some intuition into the text? Are there arny other simple activation functions that are purely nonlinear in the sense that r=0? Would x^2 work (the shape is similar to |x|)? Would x^3 (shape very different but in some sense x^3 is less linear than x^2...)? * Line 196: would it make sense to have an analogue of Figure 6 (or at least panels (a) and (b)) for linear sigma()? It should illustrate this claim directly. Maybe as a supplementary figure if there is no space in the main text? Or maybe it can be superimposed as thin lines directly into Figure 6? Or are the curves for linear sigma() somehow noninteresting? * Figure 10 in the suppl materials: the r=0 yellow line is very different from the orange line (r=0.8?), as remarked in the text. Is it really not a smooth change (line 353)? Why don't you add more lines for the values of r between orange and yellow? There is space in the figure. This is related to the word "abruptly" in line 213 of the main text. I think you need to clarify if this means "smooth but fast" or actually "non-smooth phase transition". * Figure 6 shows that bias term does not show any divergence (following Ref [19]). However, Mei & Montanari claim that the bias term does diverge in random features model: see e.g. page 6, "Bias term also exhibits a singularity at the interpolation threshold" listed as one of the main insights. This is not the primary focus of this work but it seems that Ref [19] and your results shown here are in disagreement with Mei & Montanari (possibly about how to define the bias term correctly?). As you show it here in Figure 6, please comment on this difference to Mei & Montanari. Minor comments * Title: consider removing the part after the colon. * First sentence of the abstract: it only describes "sample-wise" double descent. Maybe formulate such that it refers to both sample-wise and complexity-wise, e.g. "increasing the number of training examples N or the number of parameters P..." * A recent work on random features model: https://arxiv.org/abs/2006.05013 -- consider citing if relevant (?) The same for https://arxiv.org/abs/2002.08404. * Bullet list at line 162: mention in the text which activation functions sigma() given each of the r values. It's written in the figure but not in the text. ---------------------- POST-REBUTTAL: My score was already 8 so I leave it as it was. There was a very interesting discussion between the reviewers, that I hope the authors will get forwarded by the editor. If so, the authors could work on clarifying some things in the revised version.

[Author Response · NeurIPS 2020]

*General response*:

We wish to thank the reviewers for their valuable feedback. We are pleased that they generally appreciated the novelty of the analysis presented in this paper, which sheds light on many recent observations in the double-descent literature. We also appreciate extensive and thoughtful comments regarding clarity and pointers to relevant literature that was missing. We will implement all of them which we think will greatly improve the quality of the text, we also include more specific discussions on the main points below (**R1 R2 R3 R4** ).

We would like to start by emphasizing that the purpose of our paper is to analyze and distinguish two types of overfitting which are both attracting significant interest and are often conflated with one another. From this point of view, our paper is less about the existence of multiple peaks but more about (1) differentiating the sources and properties of the two types of peaks and (2) investigating how they interact with the nonlinearity of the activation function (which implicitly regularizes the linear peak: as suggested by **R2** we will emphasize and add more details on this).

We believe these contributions will help understand more complex cases, especially in light of recent works which also study the presence of multiple peaks. On the one hand, data distributions whose covariances have block structure can give rise to multiple *linear peaks*[1]. On the other hand, random feature regression with the NTK features of a two-layer network displays two *nonlinear* peaks due to the block structure of the covariance of the NTK features[2]. The work by Adlam and Pennington, which appeared after submission time and was mentioned by **R1** , provides an answer to **R2** 's question about the possible existence of several nonlinear peaks.

**R2** and **R3** raise valid concerns about how our picture generalizes to more complex data distributions and learning algorithms. Despite its simplicity, the setup we consider is rich enough to capture the two kinds of overfitting as desired. As far as the data distribution is concerned, we considered the unstructured iid case in the main text because it is the historical model to describe the two kinds of overfitting. Appendix C briefly discusses the case of MNIST, but the phenomenology of the linear peak becomes significantly more complex in structured datasets as illustrated by the work of Chen et. al. As for the impact of the learning algorithm, an unclear sentence pointed out by **R4** might be underselling how realistic our teacher-student framework is: by '2-layer networks', we mean 3-layer networks with 2 *hidden* layers. Furthermore, we have verified that adding extra layers has little impact on the results, a point which we will further stress.

*Specific responses:*

@**R1** **Parameter-wise vs. sample-wise**: We are not sure to understand the first weakness stated by the reviewer. Double descent was initially investigated parameter-wise but several works have subsequently studied it sample-wise. We certainly agree with the reviewer that changing the width and the dataset size is not symmetric. We vividly illustrate this fact by presenting a full visualization of the 2D phase-space, which highlights the role played by the input dimension. **'Aren't both peaks due to small eigenvalues in $Z^T Z$?'**: The linear peak is indeed related to small eigenvalues (orange curve of fig.5), but not due to strictly vanishing eigenvalues like the nonlinear peak (purple curve of fig.5). In fact, the smallest eigenvalue decreases monotonously sample-wise (for $N \leq P$) reaching zero in correspondence of the non-linear peak. An explanation for this counter-intuitive behavior is provided in appendix B. We will clarify this point.

@**R2** **Bias-variance decomposition**: We understand the concerns of the reviewer about the relevance of the section presenting a BV decomposition. The aim of this section was to highlight the fact that the linear peak is only caused by noise variance, which is why it vanishes in absence of noise in contrast to the nonlinear peak. However we acknowledge that the density of the figures may drown the message conveyed, and will consider simplifying them or moving some material to the appendix.

@**R3** **The case of MNIST**: Our contribution indeed leverages pre-existing theory, but we believe it brings along new insights and will do our best to better highlight the key takeaways. Regarding the MNIST experiment, as we will explain better, the absence of the linear peak is due to the very small intrinsic dimension of such a simple and structured dataset. A more thorough investigation of the impact of the structure of the dataset is left for future work.

@**R4** **Role of the nonlinearity**: We agree that the wording 'purely nonlinear' ($r = 0$) should be clarified. In the high-dimensional regime we focus on, the linear part of an activation function is measured by $\zeta = \mathbb{E}_{z \sim \mathcal{N}(0,1)}[z\sigma(z)]$, which vanishes for even functions such as $\sigma(x) = |x|$ but also $\sigma(x) = x^2$. The non-smoothness of generalization behavior around the point of pure linearity ($r = 1$) is also an interesting question which we will expand in SM. As for the divergence of the bias in Mei and Montanari's work[3], it is due to the fact that the bias still contains the diverging initialization variance. The latter was disentangled to yield a well-behaved bias[4].

## Footnotes

[1]L. Chen, Y. Min, M. Belkin, and A. Karbasi. Multiple descent: Design your own generalization curve.

[2]B. Adlam, J. Pennington. The neural tangent kernel in high dimensions: Triple descent and a multi-scale theory of generalization.

[3]S. Mei and A. Montanari. The generalization error of random features regression: Precise asymptotics and double descent curve.

[4]S. d'Ascoli, M. Refinetti, G. Biroli, and F. Krzakala. Double trouble in double descent: Bias and variance(s) in the lazy regime.


[Meta-Review · NeurIPS 2020]

The paper discusses the existence of a triple descent phenomenon in the test loss as a function of the dataset size. The reviewers unanimously appreciated the conceptual novelty to the paper where authors separate the two potential phenomena causing non-monotonic test error behavior in terms of number of samples. This is very relevant work for the conference and as such the reviewers have provided extensive feedback. I urge the authors to take into account the detailed feedback in their revision. Additionally, below is the anonymized transcript of some interesting discussion points which I believe highlight some confusions in the paper and I strongly encourage the authors to address them. Most importantly among these please address with a mathematical proof/extensive empirical evidence the following concern raised by R1 regarding one of the main claims in the paper: The claim that the linear peak is exhibited only in the presence of noise as such is not justified in the paper (the authors cite [6] but [6] is only for linear models), I believe with non-linear RF models, there might still be variance terms from initialization and training data, in other words, it is not clear if the total variance can exhibit a linear peak even when SNR=\inf (no noise). In addition, following R2&R3’s suggestion, I would highly recommend adding illustrative experiments and discussion that demonstrates the presence or absence of the proposed phenomenon in practical networks. Select quotes from the discussion. ------ R1 I’m also confused by one of the paper’s main contributions, that the linear peak is solely due to overfitting noise. From playing with the RF model myself, it seems that the total variance (i.e. not loss) can exhibit a linear peak even when SNR=\inf, which seems to cut against their conclusion. AC (based on quick reading) 1. The definition of linear & non-linear peak in lines 38-42 is confusing - reading the line literally, they define the linear peak only for linear models. But my understanding is that they are not focusing on linear models at all instead they consider only non-linear models and denote the phenomenon at N=D as linear peak and N=P as non-linear peak where D is the input dimensions and P is the interpolation threshold (min number of samples which can interpolate any y values on the training dataset) - I immediately see the significance of P, but I don't know what is the real significance of D? ---- Is the N=D peak a consequence of when the network behaves like a "linear" model (i.e. \hat{f}(x)= < x,w > for some w)? ---- or is it the consequence of the fact the the ground truth is linear (i.e., f^*(x) = < x,\beta > +\epsilon)? ---- (may be out of scope but) In the definition of r which seems to be crucial, the properties of nonlinearity around 0 are considered, I don't see why 0? How will the analysis change if we allowed a bias term? ---- What are \psi and \phi in eq 4 used for? Beyond [19], where else is the asymptotic in 4 used? 2. I think another major concern with the paper is discussion of related work and stating relevant results. This also makes me confused about separating what is rigorous and what is speculative in terms of the explanation they provide. Specifically, - The claim for non-linear models that the N=D peak is implicitly regularized by nonlinearity - I couldn't figure what it means and how they justify it? Can anyone clarify? - The decomposition/attributions of different variance terms to different peaks is super confusing for me. Sec 2.3 is I think important but is poorly explained. The claim is that the linear peak N=D is purely due to noise \epsilon variance. This is not obvious to me why -- they cite [6] but [6] is only for linear model, but it is not clear to me why for non-linear models the initialization variance should not affect the variance at D (with say ReLU activation or even linear activation but where the first layer is fixed overcomplete representation with P>>D). Overall, in combination with not understanding what they call as implicit regularization from non-linearity, I dont really get what the phenomenon happening at N=D is? Although they cite [19] I think to be self contained the components of the variance should be written out clearly here and they should mention what computation of this they plot in Fig 6. - Minor- In L146++ they mention [6] for showing how small non-zero eigenvalues are bad for generalization but 6 only studies linear models. Also they mention "the norm of the interpolator needs to become very large to fit small eigenvalues according to [3]" -- this is true but only for vanishing \gamma (which is the setting they work in for most parts but they do not clarify this upfront. R1 I think I can answer some of your questions. 1. ---My understanding is that the linear peak and nonlinear peak are defined through their locations, i.e. N=D and N=P. The reason for the terminology is that to analyze the RF model, one projects the activation function into a linear and nonlinear component----more specifically, into the first two Hermite polynomials and their remainder, i.e. into a linear function and a function that has been called "purely nonlinear." The constants eta and zeta govern the weights of these two components. When \zeta=\eta, the activation function is linear yielding a model of the form a*<\Theta, x>. When \zeta=0, the linear component of the activation function is 0. After this projection, one can see that these peaks coincide with different rank constraints on the kernel, the size of each of which is related to the constants \eta and \zeta. ---Adding biases to the random feature model significantly complicates the analysis. Rather than the self-consistent equation, which determines the asymptotics, being a simple polynomial it is a couple integral equation. See Adlam et al. 2019 arxiv: 1912.00827. ---The constants \psi and \phi are crucial to the model. They determine the high-dimensionality of the data and parameterization of the model. The results for the RMT problem are all asymptotic, as N,D,P -> \infty. The constants \psi and \phi fix their ratios in the limit, and changing them will alter the limit behavior. When the authors plot a theoretical prediction as a function of N, they are adjusting the constants \psi and \phi. 2. ---Previous work has noted that the spectral properties of the RF model are equivalent (in some sense) to a different matrix model, see Eq. (6). Since the linear peak ultimately comes from the term with the \sqrt{\zeta} prefactor, reducing this constant reduces the size of the linear peak. I believe this is what the authors mean by implicit regularization. ---I agree that the bias-variance decomposition is confusing, but this is not the authors' fault. There has been a huge proliferation of different bias-variance decompositions recently, and it has generated a lot of confusion. The bias-variance decomposition the authors use includes the randomness from three sources: initialization of the random features, sampling of the training points, and label noise. They then apply the law of total variance twice to the variance term, which produces three terms that they attribute to initialization of the random features, sampling of the training points, and label noise. One can argue whether the interpretation of these terms is correct. My main concern is that while the test loss does not have a peak in the SNR=\inf case (i.e. no label noise), I believe the total variance can. This cuts against one of the main claims of the paper, that the linear peak is due to label noise. I agree that there is still variance due to the initialization of the random features (see blue curves in Fig. 6), but I do not think it has a peak at N=D, so in that sense it does not cause the linear peak. I think if the authors included a mathematical statement of the claim that the linear peak is due to label noise, it would help clarify our discussion.